# Mechanisms of Resistance to Chemotherapy in Hypopharyngeal Carcinoma

**DOI:** 10.3390/biomedicines13102485

**Published:** 2025-10-12

**Authors:** Zhaoyue Lu, Zhiwei Qiang, Wenbin Lei, Zhimou Cai

**Affiliations:** Department of Otolaryngology, The First Affiliated Hospital of Sun Yat-sen University, Guangzhou 510080, China; luzhy36@mail2.sysu.edu.cn (Z.L.); qiangzhw@mail2.sysu.edu.cn (Z.Q.)

**Keywords:** hypopharyngeal carcinoma, chemoresistance, mechanism of action

## Abstract

Hypopharyngeal carcinoma (HPC) represents the most prognostically unfavorable subtype among head and neck malignancies. Platinum-based neoadjuvant chemotherapy serves as a critical therapeutic approach for improving outcomes in hypopharyngeal carcinoma; however, its efficacy remains suboptimal due to the high incidence of chemoresistance. Current research on chemoresistance has primarily focused on head and neck squamous cell carcinoma (HNSCC), yet significant heterogeneity exists between hypopharyngeal carcinoma and other head and neck tumors, limiting the direct applicability of broader HNSCC research findings to hypopharyngeal carcinoma. This review systematically summarizes recent advances in understanding the mechanisms underlying chemoresistance in hypopharyngeal carcinoma, with emphasis on the comprehensive elucidation of key mechanisms, including apoptosis evasion, enhanced DNA damage repair, augmented autophagy, and increased drug efflux. Moreover, three noteworthy special scenarios involving cancer stem cells (CSCs), epithelial–mesenchymal transition (EMT), and cancer-associated fibroblasts (CAFs) are discussed. These entities not only intrinsically participate in multiple chemoresistance mechanisms but also interact synergistically, thereby further exacerbating chemoresistance in hypopharyngeal carcinoma.

## 1. Introduction

Malignancies originating in the hypopharynx are collectively termed hypopharyngeal carcinoma (HPC), with an annual incidence rate of 0.17 to 0.8 per 100,000 individuals. HPC accounts for 0.15% to 0.5% of all systemic malignancies and 1.4% to 5% of head and neck cancers (HNCs) [1]. This disease predominantly affects males, with the pyriform sinus and posterior pharyngeal wall identified as the most common subsites, while females more frequently present with postcricoid region involvement. Hypopharyngeal squamous cell carcinoma (HPSCC) constitutes the predominant histopathological type, representing over 95% of HPC cases, while adenocarcinoma and malignant lymphoma are rare [2]. The precise etiology of HPC remains unclear, though established risk factors for HNC, including alcohol and tobacco consumption, genetic predisposition, dietary habits, nutritional deficiencies, and socioeconomic status, have been strongly associated with HPC development [3]. Characterized by high invasiveness, early metastasis, and elevated recurrence rates, HPC demonstrates extreme malignancy and represents the subtype with the worst prognosis among HNCs [4]. No definitive first-line treatment for HPC has been established, with current management primarily involving multimodal approaches incorporating surgery, radiotherapy, and chemotherapy. Platinum-based chemotherapy serves as a cornerstone for improving HPC outcomes, administered through various strategies including induction chemotherapy, concurrent chemoradiotherapy, and palliative chemotherapy. Commonly utilized agents include cisplatin, 5-fluorouracil, and taxanes [5].

Chemotherapy resistance refers to the phenomenon wherein tumor cells exhibit diminished or absent responsiveness to chemotherapeutic agents, leading to significantly reduced treatment efficacy or complete therapeutic failure. It can be categorized into intrinsic and acquired resistance. Intrinsic resistance arises from inherent genetic or epigenetic characteristics of tumor cells, such as overexpression of multidrug resistance genes (MDR1) or aberrant activation of DNA repair pathways. In contrast, acquired resistance develops progressively during chemotherapy and may involve mechanisms such as upregulation of drug efflux pumps (e.g., P-glycoprotein), suppression of apoptosis pathways, or alterations in the tumor microenvironment [6]. Chemoresistance not only reduces objective response rates (ORRs) and disease control rates (DCRs) but also significantly increases the risk of recurrence and metastasis, adversely impacting patient survival outcomes. Studies have indicated that HPC exhibits inherently low sensitivity to chemotherapeutic agents and is prone to developing chemoresistance during treatment [7]. To date, the precise mechanisms underlying chemotherapy resistance in HPC remain poorly understood, with no effective solutions currently available, contributing to the dismally poor prognosis of this patient population. Following first-line treatment, the 5-year recurrence rate of HPC reaches 41% [8], with a 5-year survival rate of only 30–35% [9]. A recent retrospective study reported an even lower 5-year survival rate of 24% for HPC, far below the average for other HNCs [10]. Currently, due to the lack of reliable biomarkers for identifying chemotherapy resistance, clinical evaluation of HPC treatment efficacy and resistance levels primarily relies on morphological changes in target lesions, as per the Response Evaluation Criteria in Solid Tumors (RECIST) guidelines [11]. This approach inevitably introduces a time lag in resistance detection, delaying therapeutic adjustments and further compromising 5-year survival rates. While extensive research has been conducted on chemotherapy resistance in HNSCC, many studies inappropriately group HPSCC with other HNSCC subtypes. Notably, HPSCC demonstrates distinct clinical or pathological features, including higher p53 mutation rates, greater invasiveness, elevated recurrence rates, and earlier metastasis. Unfortunately, as the HNC subtype with the worst prognosis, HPC remains understudied regarding both resistance mechanisms and potential solutions. To systematically review the mechanisms of chemotherapy resistance in hypopharyngeal cancer (HPC), we conducted a literature search using combinations of the following keywords: “hypopharyngeal cancer”, “FaDu cells”, “chemotherapy resistance”, “cisplatin resistance”, and “head and neck squamous cell carcinoma (HNSCC)”. Only studies involving FaDu cells or patient samples of hypopharyngeal cancer were included when considering HNSCC-related articles for mechanistic discussion. We mainly focused on recent advances from the past ten years regarding HPC chemotherapy resistance, while the older literature (>10 years) was cited when addressing classical biological processes or specific aspects of hypopharyngeal cancer.

This review synthesizes current evidence on HPC chemotherapy resistance, identifying four predominant mechanisms: (1) apoptosis evasion, (2) enhanced DNA damage repair, (3) augmented autophagy, and (4) increased drug efflux. In addition, the roles of three distinct biological contexts are explored, including cancer stem cells (CSCs), epithelial–mesenchymal transition (EMT), and cancer-associated fibroblasts (CAFs), proposing future research directions by integrating findings from HNSCC and other malignancies. 

## 2. Common Chemotherapeutic Agents for Hypopharyngeal Carcinoma and Their Mechanisms of Action

Cisplatin is the cornerstone chemotherapeutic agent for hypopharyngeal carcinoma. The National Comprehensive Cancer Network (NCCN) guidelines recommend it as a first-line treatment in various chemotherapy regimens [5]. Consequently, chemoresistance in hypopharyngeal carcinoma primarily presents as cisplatin resistance. The antitumor effects of cisplatin are predominantly mediated through the induction of DNA damage and apoptosis (Figure 1A). After entering the cell, cisplatin binds to DNA purine bases, especially guanine. It then forms intrastrand and interstrand crosslinks, which disrupt DNA replication and transcription. This results in cell cycle arrest at the G2/M phase. The ensuing DNA damage activates signaling pathways such as p53, triggering the mitochondrial intrinsic apoptotic pathway. This process upregulates pro-apoptotic proteins such as Bax, while downregulating anti-apoptotic proteins such as Bcl-2. Together, these changes activate the caspase cascade and lead to programmed cell death. In addition, cisplatin exerts antitumor effects through several alternative mechanisms. Studies have demonstrated that its antitumor efficacy may be attenuated under hypoxic conditions, underscoring the profound role of the tumor microenvironment on chemoresistance [12,13]. 

Paclitaxel and docetaxel also play pivotal roles in the chemotherapy of hypopharyngeal carcinoma and are classified as first-line agents for induction chemotherapy per NCCN guidelines [5]. Both drugs belong to the taxane class of chemotherapeutic agents and share similar mechanisms of action. They primarily target β-tubulin [14], promoting microtubule stabilization and preventing chromosomal segregation during metaphase. This induces cell cycle arrest, thereby blocking mitosis and triggering apoptosis [15]. Furthermore, these agents phosphorylate Bcl-2 and activate caspases to initiate apoptosis (Figure 1B) [16,17]. 

5-Fluorouracil (5-FU) is another key chemotherapeutic agent for hypopharyngeal carcinoma that has been designated as a first-line drug in NCCN-recommended concurrent chemoradiotherapy and induction chemotherapy regimens [5]. As a pyrimidine analog, 5-FU is metabolized inside cells into active compounds. Among them, FdUMP inhibits thymidylate synthase (TS) [18], while FdUTP disrupts DNA and RNA synthesis [19]. These actions induce apoptosis via cell cycle checkpoint activation (Figure 1C). Recent studies have also revealed that 5-FU can elicit immunogenic cell death [18]. 

Notably, all chemotherapeutic agents for hypopharyngeal carcinoma exert their anticancer effects by inducing apoptosis. This mechanism remains a key focus in strategies to overcome chemoresistance. Numerous studies are actively exploring diverse approaches to promote tumor cell apoptosis, thereby enhancing the efficacy of hypopharyngeal carcinoma chemotherapy, and ultimately reducing chemotherapeutic tolerance and improving treatment outcomes. 

## 3. Mechanisms of Chemoresistance in Hypopharyngeal Cancer

### 3.1. Apoptosis Inhibition

Apoptosis, also known as programmed cell death, is a highly regulated cellular suicide mechanism ubiquitously present in multicellular organisms, involved in tissue homeostasis, development, and immune responses [19]. Central to apoptosis is the activation of caspases, primarily regulated by the death receptor-associated extrinsic pathway and the mitochondria-associated intrinsic pathway [20]. Among these, the Bcl-2/Bax-mediated mitochondrial intrinsic apoptotic pathway plays a critical role in chemotherapy resistance in hypopharyngeal carcinoma and serves as a downstream mechanism for resistance to multiple chemotherapeutic agents. The anti-apoptotic protein Bcl-2 and the pro-apoptotic protein Bax are key apoptotic regulators in the Bcl-2 protein family, controlling apoptosis by modulating MOMP. Bcl-2 exerts its anti-apoptotic effects by inhibiting the release of apoptotic factors such as cytochrome C and suppressing Bax’s pro-apoptotic activity. In contrast, upon cellular stress, Bax transitions from a cytoplasmic monomeric state to an oligomeric form on the mitochondrial membrane, forming pores that increase MOMP, leading to the release of mitochondrial contents into the cytoplasm and subsequent cell death [21]. An increasing body of evidence suggests that in hypopharyngeal carcinoma, chemotherapeutic agents such as cisplatin activate p53 by inducing DNA damage, thereby upregulating Bax and promoting apoptosis [12,13,14,15,16,17,18,22,23]. However, studies have revealed that hypopharyngeal carcinoma cells frequently exhibit Bcl-2 overexpression and Bax downregulation, resulting in impaired apoptotic signaling and significantly enhanced cisplatin resistance. In this respect, Ma et al. [24] observed that in FaDu drug-resistant cell lines, Bcl-2 was markedly elevated, while Bax and caspases were significantly reduced, indicating pronounced apoptotic suppression in resistant cells. c-Jun, a key regulator of apoptosis, is a component of the AP-1 transcription factor and a major substrate of the JNK signaling pathway, exhibiting dual roles in apoptosis [25]. Upon phosphorylation by JNK, c-Jun upregulates FasL and Bax, thereby inducing apoptosis via the extrinsic and intrinsic pathways, respectively. However, in certain contexts, such as nasopharyngeal carcinoma, c-Jun demonstrates an opposing anti-apoptotic effect [26,27]. Liu et al. [28] found that c-Jun was significantly upregulated in hypopharyngeal carcinoma resistant to Taxel, Platin and Fluorouracil (TPF) chemotherapy, suggesting that its conventional anti-apoptotic role may not be operative. It is widely thought that c-Jun might function similarly in hypopharyngeal carcinoma as in nasopharyngeal carcinoma [29], where it upregulates Bcl-2 to inhibit apoptosis and confer Chemoresistance. However, the expression differences in Bcl-2 and Bax were not assessed, leaving this hypothesis unverified. Interestingly, it is now understood that the tumor microenvironment (TME) also influences apoptosis. Hypoxia, a hallmark of TME, arises when the oxygen demand of rapidly proliferating tumors exceeds the vascular supply [30]. Hypoxia can directly upregulate Twist1 via HIF-1α [31]. According to Lu et al. [32], in the FaDu/Twist+ hypopharyngeal carcinoma cell line, caspase-3, caspase-9, Bcl-2, and Bax exhibited alterations favoring anti-apoptosis, suggesting that Twist1 expression may promote chemotherapy resistance by suppressing apoptosis. In addition, severe hypoxia activates the unfolded protein response (UPR) pathway via HIF-1α, upregulating GRP78 and Bcl-2 while inhibiting CHOP-mediated pro-apoptotic signaling, further exacerbating chemoresistance (Figure 2A) [33]. Beyond the Bcl-2/Bax-associated intrinsic apoptotic pathway, the inhibitor of apoptosis protein (IAP) family also plays a significant role in apoptosis suppression, as extensively studied in other head and neck squamous cell carcinomas [34]. While this mechanism remains understudied in hypopharyngeal carcinoma, Zeng et al. [35] reported that overexpression of IAP family members in cisplatin-resistant cell lines is strongly correlated with chemoresistance. Moreover, alterations in redox homeostasis also play an essential role in regulating apoptosis and mediating chemoresistance. Recent studies have demonstrated that elevated TNFAIP2 stabilizes and activates NRF2 by competitively binding to KEAP1, thereby enhancing the cellular antioxidant defense system. Sustained NRF2 activation reduces the accumulation of reactive oxygen species (ROS) during chemotherapy, which in turn suppresses ROS-induced JNK activation and subsequent apoptosis, ultimately conferring pronounced chemoresistance [36].

Based on these findings, many strategies to overcome chemotherapy resistance in hypopharyngeal carcinoma target the Bcl-2/Bax-associated intrinsic pathway. Other approaches focus on its upstream regulators to enhance apoptosis and reduce chemoresistance. Therapeutic approaches can be categorized into three types: Epidermal growth factor receptor (EGFR)-targeted drugs, natural compounds, and gene therapy. 

EGFR-targeted drugs are considered a promising option for addressing hypopharyngeal carcinoma chemotherapy resistance. EGFR activates the PI3K/AKT pathway. This leads to phosphorylation and inactivation of the pro-apoptotic protein BAD, thereby blocking mitochondrial-dependent apoptosis [37]. In addition, EGFR activates the MAPK pathway to upregulate anti-apoptotic proteins such as Bcl-2 and Bcl-xL. This enhances cellular resistance to apoptotic stimuli [38]. Song et al. [39] found that anlotinib significantly increased apoptosis in hypopharyngeal carcinoma cells. Currently, anlotinib is used clinically in advanced non-small cell lung cancer (NSCLC) and soft tissue sarcoma. In these cancers, it suppresses Bcl-2 indirectly through angiogenesis inhibition or directly through the JNK pathway, thereby inducing apoptosis [40,41]. However, its application in hypopharyngeal carcinoma has yet to enter clinical trials. Combination therapies involving EGFR inhibitors are also under investigation. Kleszcz et al. [42] studied erlotinib in combination with Wnt/β-catenin and PI3K inhibitors. They found that PRI-724 (a Wnt inhibitor) combined with erlotinib or HS-173 (a PI3K inhibitor) significantly enhanced apoptosis, especially early apoptosis, in CAL 27 and FaDu cells. These results suggest potential synergy between EGFR inhibitors and Wnt/PI3K inhibitors.

Emerging evidence suggests that naturally derived compounds may offer therapeutic potential in addressing hypopharyngeal carcinoma resistance. Among these, Biochanin A, a natural isoflavone that binds estrogen receptors, exerts estrogen-like effects. Studies have indicated that biochanin A promotes apoptosis by inhibiting the Akt and MAPK pathways, reducing NF-κB activity, and modulating the Bcl-2/Bax ratio [43], demonstrating anti-cancer effects in breast and prostate cancers [44]. In-A Cho et al. [45] confirmed these effects in hypopharyngeal carcinoma, observing a 4.2-fold increase in apoptosis in FaDu cells treated with biochanin A. Moreover, FasL (a death receptor) and Bcl-2 decreased in a dose-dependent manner. This finding suggests that biochanin A activates both extrinsic and intrinsic apoptotic pathways. Such dual activation highlights its potential to overcome chemotherapy resistance. Chemically modified curcumin (L42H17) has also demonstrated significant anti-cancer potential in hypopharyngeal carcinoma. By targeting the NF-κB pathway, it downregulates Bcl-2 while upregulating pro-apoptotic proteins (e.g., cleaved-PARP, cleaved-caspase-3) to induce apoptosis [46]. Costunolide, another promising agent, blocks cell cycle progression, upregulates Bax, and downregulates Bcl-2 to promote apoptosis [47]. Li et al. [48] reported that costunolide, used alone or with cisplatin, significantly increased apoptosis in FaDu cells. The effect was mediated by ROS generation and upregulation of intrinsic apoptotic pathway-related proteins. Furthermore, costunolide was found to enhance apoptosis by suppressing the AKT and NF-κB pathways. 

Gene therapy is also considered a promising approach for overcoming hypopharyngeal carcinoma chemotherapy resistance. p53 is the most extensively studied tumor suppressor gene. It regulates the cell cycle and repairs DNA damage. It also upregulates pro-apoptotic genes to induce programmed cell death, thereby inhibiting cancer progression [49]. Ren et al. [50] demonstrated that co-expression of p53 and *ING4*, combined with cisplatin, significantly reduced tumor weight and induced apoptosis in hypopharyngeal carcinoma. However, their application remains limited to animal models due to the dose-limiting hepatotoxicity associated with adenoviral vector systems. Identifying key gene targets and optimizing vector structures remain critical challenges. Despite remarkable progress in gene therapy research, substantial hurdles remain before these approaches can be successfully implemented in clinical oncology practice. 

### 3.2. Augmented Autophagy 

Autophagy is a sophisticated intracellular degradation mechanism. It maintains cellular homeostasis and responds to external stimuli by degrading and recycling damaged or redundant components (e.g., impaired organelles, misfolded proteins, pathogens) via lysosomes [51]. The initiation of autophagy primarily involves the inhibition of the mTOR pathway or activation of the AMPK pathway under various stress conditions, leading to the phosphorylation of the ULK1 complex. Subsequently, the activated ULK1 complex recruits and activates the downstream class III PI3K complex (comprising VPS34, Beclin1, ATG14L, and p150) through phosphorylation. The elongation of autophagosomal membranes is then regulated by ATG complexes and LC3-II. Following autophagosome formation, fusion with lysosomes is mediated by SNARE proteins, resulting in the degradation of autophagic cargo (Figure 2B) [52,53]. Previous studies have demonstrated a dual role of autophagy in chemotherapy. On one hand, autophagy protects tumor cells by clearing damaged organelles and proteins, allowing them to survive chemotherapy-induced stress and contributing to chemoresistance [54]. On the other hand, under certain conditions such as apoptosis deficiency, excessive autophagy may trigger autophagic cell death and enhance chemotherapy efficacy [55]. In current research on hypopharyngeal carcinoma, autophagy predominantly functions as a protective mechanism, promoting chemoresistance in tumor cells. Lin et al. [56] revealed that cisplatin-resistant FaDu cells exhibit differentially expressed extrachromosomal circular DNA (eccDNA), among which the autophagy-inducing gene *RAB3B* is amplified by eccDNA, leading to chemoresistance via autophagy-mediated clearance of chemotherapeutic agents. This study provides novel insights into chemoresistance mechanisms, suggesting that resistance may arise not only from aberrant signaling pathways but also from dysregulated gene expression. However, research on hypopharyngeal carcinoma chemoresistance remains limited, particularly regarding gene expression abnormalities. In addition, while autophagy’s role in tumor chemoresistance has been extensively studied, direct investigations in hypopharyngeal carcinoma remain scarce. 

To address autophagy-mediated chemoresistance, several therapeutic approaches have been explored, including pharmacological autophagy inhibitors, induction of excessive autophagy to trigger autophagic cell death, modulation of autophagy-related pathways (e.g., PI3K/AKT/mTOR, MAPK), and epigenetic modulation [57,58,59]. Zhang et al. [60] reported that autophagy and the autophagy-related protein Beclin-1 were significantly enhanced in cisplatin-resistant FaDu cells. Treatment with the autophagy inhibitor 3-MA induced cell cycle arrest and markedly increased apoptosis, accompanied by reduced levels of Beclin-1 and upregulated p62 expression. These studies overlap in their assertion that autophagy inhibition significantly alleviates chemoresistance in hypopharyngeal carcinoma, offering a promising therapeutic strategy. However, this drug has not yet progressed to clinical trials. 

### 3.3. Enhanced DNA Repair

DNA damage repair represents a well-established mechanism contributing to chemotherapy resistance in hypopharyngeal carcinoma. Commonly used chemotherapeutic agents for hypopharyngeal carcinoma, such as cisplatin and 5-Fu, are now understood to induce apoptosis by damaging DNA. Cellular defense against such damage occurs through multiple repair pathways, including direct repair (DR), base excision repair (BER), nucleotide excision repair (NER), mismatch repair (MMR), and translesion synthesis (TLS) [61]. Previous studies have demonstrated that tumor cells enhance repair efficiency primarily by upregulating repair genes and activating the NHEJ/HR pathway to counteract DNA damage induced by chemotherapeutic agents [62]. Osman et al. [63] showed that cisplatin-induced oxidative stress activates NRF2 signaling, and KEAP1 inactivation mutations further amplify this pathway. NRF2 binds to antioxidant response elements (AREs) on DNA, driving continuous transcription of antioxidant and detoxification genes. This mechanism regulates intracellular ROS and glutathione levels, thereby protecting cancer cells from cisplatin-induced DNA damage and contributing to chemoresistance (Figure 2C). Although the study did not employ hypopharyngeal carcinoma cell lines, it highlighted significant mutations in NRF2 and KEAP1 in hypopharyngeal carcinoma cells, suggesting a high likelihood that these cells similarly acquire chemoresistance via the NRF2 axis. In addition, NER and its associated protein ERCC1 have been frequently reported in previous studies on head and neck squamous cell carcinoma, with strong evidence supporting their association with cisplatin resistance [64,65,66]. However, related research on hypopharyngeal carcinoma remains to be conducted. 

Current therapeutic strategies targeting enhanced DNA repair primarily focus on inhibiting DNA repair pathways and the NRF2/KEAP1 axis [67]. However, no clinical or preclinical investigations have specifically evaluated such interventions in hypopharyngeal carcinoma. The findings of Osman et al. [63] suggest that NRF2-targeted therapy may represent a promising direction for reversing chemoresistance in hypopharyngeal carcinoma. 

### 3.4. Increased Drug Efflux 

Increased drug efflux is a major mechanism of multidrug resistance (MDR). It reduces intracellular drug levels and counteracts the effects of chemotherapy. ABC transporters, including P-glycoprotein (P-gp) and multidrug resistance-associated protein 1 (MRP1), act as key efflux pumps in tumor cells (Figure 2D). These proteins remove antitumor drugs using energy from ATP hydrolysis [68]. Among them, P-gp, encoded by the *ABCB1* gene, is the most extensively studied transporter. It reduces drug accumulation in tumor cells by altering cell membrane permeability or directly effluxing drugs [69]. In addition, copper transporters represent another class of drug efflux transporters [70]. Analysis of clinical data from 317 HNSCC patients in the Cancer Genome Atlas (TCGA) database revealed that high expression of *MDR1* and *MRP1* genes was associated with cisplatin resistance and poor survival [71], indicating that drug efflux pumps play a significant role in cisplatin resistance in HNSCC patients and markedly reduce survival rates. A similar phenomenon has been observed in hypopharyngeal carcinoma, with a study by Lu et al. [32] demonstrating that the hypopharyngeal carcinoma cell line FaDu, after induction of chemoresistance by paclitaxel, exhibited elevated expression of Twist1 and MDR1/P-gp in an MDR-dependent manner, suggesting that Twist1 contributes to chemoresistance in hypopharyngeal carcinoma cells by upregulating MDR1/P-gp.

Current therapeutic approaches to overcome chemoresistance caused by drug efflux primarily include efflux pump inhibitors, nanomaterials, light-based therapies, gene editing, and drug repurposing [72]. The efficacy of efflux pump inhibitors has been reported in hypopharyngeal carcinoma. Ma et al. [73] successfully restored chemosensitivity in the resistant FaDu cell line using the proteasome inhibitor MG-132. Their study found that MG-132 could suppress P-gp by activating the JNK pathway, thereby reducing drug efflux and reversing chemoresistance in hypopharyngeal carcinoma. 

In summary, chemotherapy resistance in hypopharyngeal carcinoma involves multiple mechanisms, including apoptosis inhibition, augmented autophagy, enhanced DNA repair, and increased drug efflux. These common mechanisms are summarized in Table 1.

## 4. Cancer Stem Cells, EMT, and CAFs in Chemoresistance

Cancer stem cells (CSCs) represent a small subpopulation of tumor cells with stem-like properties. They possess self-renewal and differentiation capacities similar to normal stem cells but lack proper regulatory control [74]. CSCs are widely recognized as fundamental drivers of tumorigenesis, disease progression, and clinical recurrence, playing a profound role in resistance to chemotherapy and radiotherapy [75]. Current evidence suggests that CSCs integrate multiple chemoresistance mechanisms and demonstrate extreme chemoresistance due to their plasticity and quiescent state [76]. Established CSC chemoresistance mechanisms include: (1) drug efflux [77,78], (2) apoptosis evasion [79], (3) enhanced DNA damage repair [78], and (4) dysregulation of core stemness pathways such as Hippo/YAP1, Wnt/β-catenin, Hedgehog, Notch, and JAK/STAT signaling networks [80,81].

In the context of hypopharyngeal carcinoma, our research team previously identified a CD271-high-expressing CSC subpopulation with superior proliferative and tumorigenic capacities, with IL-6 overexpression implicated as a chemoresistance mechanism [82]. Imai et al. [83] observed that CD271^+^ tumor cells increased from 16.3% to 35.2% post-cisplatin treatment, demonstrating marked cisplatin resistance mediated by upregulation of ABC transporter expression, specifically ABCC2 (2.5-fold), ABCB5 (4.8-fold), and ABCG2 (2.4-fold), establishing a mechanistic association between this CSC subpopulation and active drug efflux-mediated resistance.

The role of cyclooxygenase-2 (COX-2) in hypopharyngeal CSC resistance was investigated by Saito et al., who reported an inverse correlation between COX-2 expression in HNSCC specimens and pathological response to induction chemotherapy. In addition, ex vivo inhibition of COX-2 reduced the half-maximal inhibitory concentration (IC_50_) of docetaxel in FaDu cells but not in Detroit 562 cells, concurrently downregulating stemness markers (OCT3/4, NANOG, SOX-2, ALDH1A1) and impairing sphere formation. These findings collectively suggest that COX-2 promotes chemoresistance by enhancing CSC properties, thereby identifying it as a potential therapeutic target [84]. Current research into hypopharyngeal CSCs is limited by factors such as tumor heterogeneity and individual variability. Proposed CSC-directed therapies include PRI-724 + vismodegib combination (Kleszcz et al. [42]) and tunicamycin-induced endoplasmic reticulum stress to disrupt integrin β1/TGF-β1/Smad signaling (Gu et al. [85]), though neither has advanced to clinical trials.

Epithelial–mesenchymal transition (EMT) is a process in which epithelial cells lose polarity and tight junctions while acquiring mesenchymal traits, including migration and invasion. Regulated by TGF-β, Wnt, and Notch pathways via Snail, Twist, and ZEB transcription factors [86], EMT facilitates embryonic development, tissue healing, and cancer progression, particularly metastasis and chemoresistance [87]. In the context of cancer, EMT often manifests as a hybrid “partial EMT” (pEMT) state, where cells co-express both mesenchymal (vimentin) and epithelial (E-cadherin) markers [86], which enhances stemness and therapy resistance [87]. Our single-cell RNA sequencing revealed the presence of both complete and pEMT transitions in hypopharyngeal carcinoma [88], where EMT confers resistance via multidrug resistance protein activation, proliferative suppression, and apoptotic blockade [87]. EMT transcription factors (SNAI1/2, ZEB1) additionally mediate DNA damage repair [89]. While the EMT-chemoresistance relationship has been extensively studied in HNSCC, hypopharyngeal-specific data remain limited. CD147, known as EMMPRIN, is a molecule that promotes EMT. Studies have demonstrated that its interaction with cyclophilin A (CyPA) enhances the chemoresistance of FaDu cells [90]. Research by Zeng et al. [35] revealed that downregulating S100A9 led to the upregulation of E-cadherin in HPC cells and reversed their cisplatin resistance, suggesting that S100A9 induces chemotherapy tolerance in tumor cells by promoting EMT. Therapeutic studies further support the role of EMT in chemoresistance of HPC. For instance, Wang et al. [91] found that downregulating RBM17 not only reversed the EMT phenotype in HPC cells but also effectively restored their sensitivity to cisplatin. Thus, existing research provides compelling evidence for the potential role of EMT in chemoresistance in HPC, though direct evidence specific to this malignancy remains limited.

Given the extensive research on HNSCC and the conserved mechanisms of EMT, it is widely thought that EMT plays a significant role in chemoresistance in HPC. Future studies should focus on HPC-specific EMT mechanisms and explore EMT-related targets as potential therapeutic strategies to overcome chemotherapy resistance [92]. 

The tumor microenvironment (TME) comprises CAFs, immune cells, endothelial cells, and extracellular matrix. It evolves under therapeutic pressure, fueling malignancy and resistance. Among these stromal components, CAFs exert profound regulatory roles in TME [93]. Historically overlooked in cancer therapies targeting tumor cells alone, CAFs have recently emerged as key chemoresistance mediators [94,95]. Kinugasa et al. [96] reported that CAF-expressed CD44 maintains CSC populations in hypoxic HNSCC niches, while CAF-secreted TGF-β/IL-6 activates PI3K/AKT/mTOR or NF-κB to enhance cisplatin resistance [97]. Our single-cell analysis identified fibroblast activation protein (FAP)^+^ myCAFs as the dominant CAF subtype in HPC, engaging in TGF-β/Notch/Wnt-mediated crosstalk with tumor cells, pathways implicated in chemoresistance [88]. Huang et al. [98] demonstrated that CAF-derived small extracellular vesicles (sEVs) induce EMT by upregulating N-cadherin/vimentin, and downregulating E-cadherin, and extracellular matrix (ECM) stiffening via fibronectin, limiting drug penetration. Despite these advances, CAF-targeted strategies for hypopharyngeal carcinoma remain largely unexplored.

Collectively, CSCs, EMT, and CAFs represent additional mechanisms contributing to chemotherapy resistance in hypopharyngeal carcinoma, complementing the classical pathways. These mechanisms are summarized in Table 2.

## 5. Outlook

Early symptoms of HPC are often inconspicuous. As a result, most patients are diagnosed at advanced stages. This leads to limited treatment opportunities and a narrow therapeutic window. In addition, the high rate of chemotherapy resistance in HPC significantly compromises patient quality of life and prognosis. Consequently, there is an urgent clinical need for well-defined biomarkers to guide HPC treatment and improve patient outcomes. Human papillomavirus (HPV) infection status has recently attracted significant interest in HNSCC research. Studies have demonstrated that HPV-positive (HPV+) patients exhibit superior chemotherapy response rates [99]. However, the HPV infection rate in HPC is relatively low (ranging from approximately 17.7% to 23.9%) [100,101], which is significantly lower than in oropharyngeal cancer. Therefore, HPV status and its closely associated biomarker p16 may not serve as optimal predictors of treatment efficacy in HPC [102]. In addition, p53 is a critical tumor suppressor gene that presents another potential biomarker. Mutations in p53 impair cellular responses to DNA damage, ultimately enhancing chemotherapy resistance in cancer cells [103]. Given the relatively high incidence of p53 mutations in HPC, p53 may serve as a promising biomarker for predicting chemotherapy response. Emerging molecular targets are also under investigation. Yumiko et al. identified four genes (*AGR2*, *PDE4D*, *NINJ2*, and *CDC25B*) significantly associated with chemoresistance in HPC cells [104], suggesting their potential as predictive biomarkers warranting further in vivo validation. Tan et al. [105] compared transcriptomic data from locally advanced laryngeal and hypopharyngeal cancer tissues with varying responses to TPF induction chemotherapy and constructed a six-gene model (*NRIP1*, *GIMAP7*, *CD72*, *THBS4*, *ABCA9*, and *SNED1*) to predict chemotherapeutic response. To date, this model represents the most comprehensive and well-validated biomarker study for HPC chemotherapy efficacy. Thus, further mechanistic studies and multi-institutional clinical trials are essential to elucidate the clinical utility of these identified biomarkers and their impact on patient outcomes. 

Beyond the identification of biomarkers, therapeutic advancements in HPC remain a critical research focus. Long-term follow-up data from the TAX324 trial [106], based on a subgroup analysis of 166 HPC and laryngeal cancer patients, demonstrated that the TPF regimen outperformed PF in progression-free survival (PFS), with significantly lower disease progression risk in the TPF group. Clinical guidelines classify HPC as a distinct subsite of head and neck cancer. However, most studies, including TAX324, do not analyze HPC separately. Instead, they group it with laryngeal and oropharyngeal cancers. This practice hinders HPC-specific therapeutic progress. Furthermore, due to its low occurrence rate and poor prognosis, HPC receives limited clinical attention, impeding large-scale clinical trials. To address chemotherapy resistance in HPC, future research should explore regimen optimization, chemosensitization, targeted therapy, and immunotherapy. Immunotherapy enhances the host immune system to recognize and attack cancer cells, primarily through immune checkpoint blockers (ICBs), CAR-T cell therapy, and cancer vaccines. ICB therapy has rapidly advanced in recent years. Randomized trials revealed that the PD-L1 inhibitor pembrolizumab monotherapy significantly improved overall survival in patients with a combined positive score (CPS) > 1 [107]. Currently, pembrolizumab is approved for recurrent/metastatic head and neck cancer with CPS ≥ 1. A clinical study conducted by our research team demonstrated that pembrolizumab combined with neoadjuvant chemotherapy achieved a higher ORR than conventional neoadjuvant chemotherapy in HPSCC [108]. Pembrolizumab improves survival in melanoma and non-small cell lung cancer. In contrast, its objective response rate (ORR) in HPC is limited [107]. This may be due to the high prevalence of PD-L1-low tumors [108]. In addition, the immunosuppressive TME in HPC dampens T-cell activity [95]. Advances in genetic engineering offer new prospects for cancer immunotherapy. PD1-adjuvant-engineered exosomes have been shown in murine models to promote dendritic cell maturation and reverse CD8^+^ T-cell exhaustion in breast cancer and melanoma [109]. Thus, reconstructing immune function via genetic engineering or nanotechnology may represent a novel strategy for HPC treatment. Targeted therapy functions by inhibiting tumor growth and metastasis by specific blockade of molecular pathways (e.g., growth factor receptors). EGFR inhibitors are the primary targeted agents for HNSCC. Cetuximab, the first Food and Drug Administration (FDA)-approved targeted drug for HNSCC, is now recommended by NCCN guidelines. However, resistance to cetuximab remains a prevalent challenge, with only 13% of HNSCC patients responding [110]. Studies have shown that multiple resistance pathways are activated during cetuximab treatment [111,112]. Moreover, cetuximab may recruit immunosuppressive Tregs. This recruitment further impairs antitumor immunity [113]. Combining targeted therapy with immunotherapy may thus overcome resistance. To combat chemotherapy resistance and improve patient survival, deeper mechanistic investigations and clinical trials on novel treatment strategies are imperative. 

## 6. Conclusions

HPC acquires chemoresistance primarily through mechanisms such as apoptosis evasion, enhanced DNA damage repair, augmented autophagy, and increased drug efflux. In addition, the complex interplay among EMT, CAFs, and CSCs integrates multiple resistance mechanisms, further exacerbating chemoresistance in HPC cells. It must be emphasized that chemoresistance in HPC has received limited attention and is often grouped with HNSCC in studies, resulting in a paucity of clinical trial data specifically designed for HPC patients. A prevailing consensus indicates that single-modality therapy is only indicated for early-stage HPC (T1 and select T2 lesions without lymph node metastasis), whereas multimodal therapy is indicated for advanced-stage patients. Notably, a significant discrepancy exists between preclinically validated therapeutic targets and the actual efficacy of candidate molecules, with existing evidence suggesting that these drugs achieve only partial target inhibition at clinically tolerable doses. Furthermore, combining agents targeting different pathways often leads to intolerable toxicity. Consequently, even when cost considerations are set aside, multi-drug regimens are frequently impractical. Future therapeutic directions for HPC will continue to focus on genetic engineering technologies, immunotherapy, and precision radiotherapy techniques. Given its rarity yet dismal prognosis, HPC requires increased attention from clinicians and researchers. Overcoming chemoresistance is crucial to improving the quality of life and survival outcomes for this patient population.

## Figures and Tables

**Figure 1 biomedicines-13-02485-f001:**
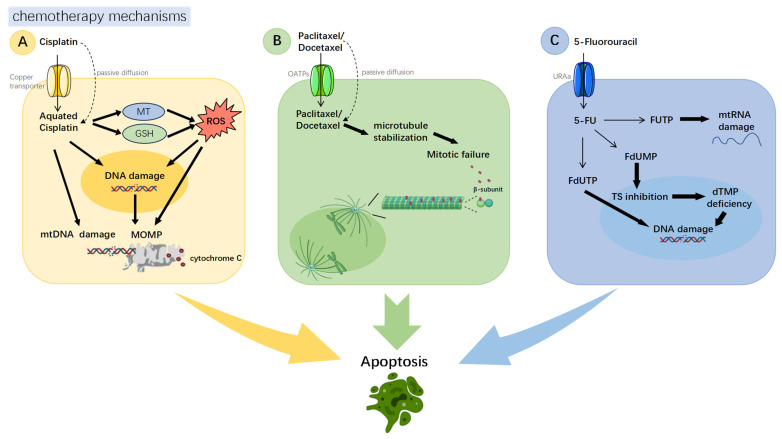
**Mechanisms of chemotherapeutic agents.** (**A**) Cisplatin: Cellular uptake occurs primarily via copper transporters and is aquated in the cytosol. The aquated form of cisplatin induces DNA damage by binding to nuclear or mitochondrial DNA. Concurrently, aquated cisplatin also interacts with cytoplasmic components like reduced glutathione (GSH) and metallothionein (MT), which results in the generation of reactive oxygen species (ROS) that trigger mitochondrial outer membrane permeabilization (MOMP). Both DNA damage and MOMP lead to cell death. (**B**) Taxanes (Docetaxel/Paclitaxel): These agents bind β-tubulin within microtubules, stabilizing polymerized tubulin. This stabilization suppresses microtubule dynamics, arrests mitotic progression, and ultimately results in cell death. (**C**) 5-Fluorouracil (5-FU): Entering cells via uracil transporters due to its uracil-like analog structure, intracellular 5-FU is metabolized into three primary active species that impair nucleic acid function: (i) Fluorodeoxyuridine monophosphate (FdUMP) potently inhibits thymidylate synthase (TS); (ii) Fluorodeoxyuridine triphosphate (FdUTP) causes misincorporation into DNA; (iii) Fluorouridine triphosphate (FUTP) is incorporated into RNA, disrupting normal function.

**Figure 2 biomedicines-13-02485-f002:**
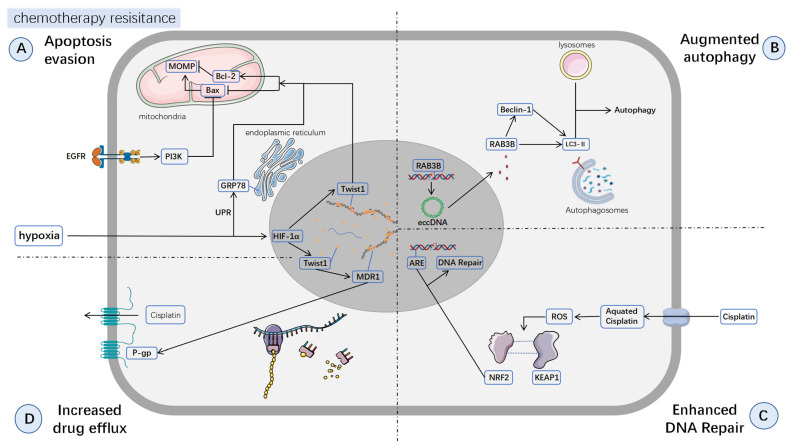
**Molecular mechanisms of chemotherapy resistance.** The schematic summarizes four key resistance pathways in hypopharyngeal carcinoma: (**A**) apoptosis evasion, (**B**) augmented autophagy, (**C**) increased drug efflux, and (**D**) enhanced DNA repair.

**Table 1 biomedicines-13-02485-t001:** Common mechanisms of chemotherapy resistance in hypopharyngeal cancer.

Type	Sort	Cancer	Marker/Pathway	Drug	Mechanisms	Action	Ref.
Apoptosis evasion	O	HPC	C-Jun	Docetaxel Cisplatin 5-Fu	TPF induces c-Jun upregulation SAG-SCF E3 ligase inhibits c-Jun degradation c-Jun resists apoptosis and regulates cell cycle (in NPC)	-	[28,29]
		HPC	Twist1	paclitaxel	Microenvironment hypoxia leads to Twist1 upregulation. Twist1 regulates Bax and Bcl-2 to resist apoptosis	-	[32]
		HNSC	GRP78 CHOP HIF-1α	Cisplatin	Severe hypoxia induces UPR, leading to GRP78 upregulation GRP78 regulates Bax and Bcl-2 to resist apoptosis	-	[33]
		HNSC (sample includes HPC)	TNFAIP2 KEAP1 NRF2 JNK pathway	Cisplatin	High TNFAIP2 stabilizes NRF2 by competitively binding KEAP1, enhances antioxidant defense, reduces ROS-induced JNK activation, and suppresses apoptosis, thereby promoting chemoresistance.		[36]
	T	HPC	EGFR	Cisplatin	EGFR activates the PI3K pathway PI3K pathway inactivates BAD, blocking apoptosis EGFR activates PI3K pathway to downregulate BAD, blocking apoptosis	Anlotinib	[39]
		HNSC (The sample was taken from HPC)	Wnt pathway EGFR PI3K pathway		Wnt pathway/β-catenin upregulates anti-apoptotic gene expression	PRI-724 and Erlotinib	[42]
		HPC	Bcl-2 FasL		Biochanin-A regulates FasL, reducing Bcl-2	Biochanin-A	[45]
		HPC	Bcl-2 Bax		L42H17 reduces Bcl-2 and increases Bax	L42H17	[46]
		HPC	ROS AKT pathway NFκB pathway	Cisplatin	CTL induces reactive oxygen species (ROS) production CTL inhibits AKT and NFκB pathways to induce apoptosis CTL regulates Bcl-2 and Bax to induce apoptosis	CTL	[48]
Augmentedautophagy	O	HPC	*RAB3B* eccDNA	Cisplatin	RAB3B induces cellular autophagy eccDNA amplification increases RAB3B gene copy, enhancing autophagy	-	[56]
	T	HPC	Beclin-1	Cisplatin	3-MA disrupts the formation of the Beclin-1 complex	3—MA	[60]
Enhanced DNA Repair	O	HNSC (mention HPC)	NRF2 KEAP1	Cisplatin	Cisplatin induces ROS production ROS disrupts the binding of NRF2 to KEAP1 NRF2 binds to ARE, initiating DNA repair	-	[63]
Increased drug efflux	T	HPC	Twist1 P-gp	Taxel	Microenvironment hypoxia leads to Twist1 upregulation. Twist1 regulates P-gp to increase drug efflux	-	[33]
		HPC	MG-132 P-gp JNK pathway	Taxel	MG-132 downregulates P-gp expression by activating the JNK pathway	MG-132	[73]

O: Mechanism of chemoresistance occurrence, T: Mechanism of chemoresistance therapy. NPC: Nasopharyngeal carcinoma; 5-Fu: 5-Fluorouracil; TPF: Taxane, Platinum, Fluoropyrimidine (combination chemotherapy regimen); GRP78: 78 kDa Glucose-Regulated Protein; EGFR: Epidermal Growth Factor Receptor; PRI-724: a CBP/β-catenin inhibitor; FasL: Fas Ligand; CTL: costunolide; eccDNA: Extrachromosomal Circular DNA; MDR1: Multidrug Resistance Protein 1; L42H17: Chemically modified curcumin; ARE: Antioxidant Response Element.

**Table 2 biomedicines-13-02485-t002:** Cancer Stem Cells, EMT, and CAFs in Chemoresistance of hypopharyngeal cancer.

Type	Sort	Cancer	Markers	Mechanisms	Ref.
CSCs	O	HPC	CD271 ATP-Binding Cassette Transporter IL-6	Increased expression of ABC transporter proteins and IL-6 in CD271+ cells.	[82,83]
		HPC	COX-2	COX-2 suppresses the expression of CSC-related genes while reducing cellular drug tolerance	[84]
	T	HPC	TGF-β1/Smad	TM inhibits the maturation of integrin β1 (130-kD), disrupts the integrin β1/TGF-β1/Smad signaling pathway, reduces TGF-β1 secretion, and undermines the microenvironmental support for CSCs.	[87]
		HPC	Wnt/Hh pathway	The Wnt/Hh pathway promotes the expression of CSC-related genes.	[42]
EMT	O	HPC	CD147	CD147 significantly upregulates MMP-9 through interaction with cyclosporin A, enhancing chemoresistance.	[94]
		HNSC (mention HPC)	KEAP1 NRF2	Hyperactivation of the KEAP1/NRF2 signaling pathway upon CDDP exposure not only induces CDDP resistance but also enhances the metastatic phenotype in HNSCC.	[63]
		HPC	S100A9	S100A9 promotes EMT and contributes to cellular chemoresistance.	[35]
	T	HPC	RBM17	Knockdown of RBM17 reduces Vimentin and ZEB1 expression, increases E-cadherin expression, significantly enhances cisplatin’s inhibitory effect on FaDu cells, and reduces cell viability.	[91]
CAFs	O	HPC	Notch pathway Wnt pathway TGF-βpathway	Interactions between metastatic cancer-2 cells and mCAFs are the most frequent, with ligand-receptor pairs enriched in various cancer-related pathways, including Notch, Wnt, and TGF-β pathways.	[88]
		HNSC (The sample was taken from larynx and hypopharynx)	sEV TGF-β1	Fibronectin is significantly upregulated in NFs following sEV-TGFβ1 treatment. Fibronectin promotes ECM assembly by CAFs, enhances tumor cell invasion, and is associated with poor prognosis in HNSCC patients.	[98]

ABCB5: ATP-Binding Cassette Subfamily B Member 5; ABCG2: ATP-Binding Cassette Subfamily G Member 2; BCRP: Breast Cancer Resistance Protein; RBM17: RNA Binding Motif Protein 17; ECM: Extracellular Matrix; IC50: Half Maximal Inhibitory Concentration.

## Data Availability

No new data were created or analyzed in this study. Data sharing is not applicable to this article.

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
