# Peer review of "Mechanisms of Resistance to Chemotherapy in Hypopharyngeal Carcinoma"

_biomedicines, 2025, doi:10.3390/biomedicines13102485_

Round 1

Reviewer 1 Report

Comments and Suggestions for Authors

This review addresses chemoresistance in HPC, a head and neck cancer subtype with particularly poor prognosis. The authors note that most research on chemoresistance is generalized to HNSCC, which may not fully capture the unique biology of HPC. The review focuses on four major mechanisms (apoptosis evasion, enhanced DNA damage repair, autophagy, and drug efflux) and highlights the roles of CSCs, EMT, and CAFs as key contributors that synergistically worsen chemoresistance.

While the manuscript is promising, there is room for improvement in several areas. Please refer to the points detailed below:

# The manuscript is generally well-written, though in some sections the sentences are quite long and complex. Breaking them into shorter, more concise statements would improve readability and flow.

# Ensure terms like “chemoresistance,” “drug resistance,” and “treatment resistance” are used consistently throughout the manuscript.

# While the manuscript cites important foundational studies, many references are over a decade old. The authors are encouraged to update the reference list by incorporating more recent literature (from the past 5–6 years) to strengthen the review and ensure it reflects the current state of research.

Author Response

  1. The manuscript is generally well-written, though in some sections the sentences are quite long and complex. Breaking them into shorter, more concise statements would improve readability and flow.

Response: Thank you for carefully reviewing our manuscript and providing valuable suggestions. We fully agree with your opinion. Regarding this point, we have thoroughly reorganized and revised the entire text. We have broken down the complex long sentences into multiple shorter ones and simplified the sentence structure to make it easier for readers to understand and read. We believe that after these revisions, the readability and logical coherence of the manuscript have been significantly improved. Once again, we thank you for helping us enhance the quality of the manuscript.

  1. Ensure terms like “chemoresistance,” “drug resistance,” and “treatment resistance” are used consistently throughout the manuscript.

Response: Thank you for pointing out the inconsistency in the use of terms in our manuscript. We fully agree with your opinion that maintaining consistency in terms is crucial for the clarity and professionalism of the article. Based on your suggestion, we have conducted a systematic review of the entire text and have uniformly changed all instances of "drug resistance" and "treatment resistance" to the more specific and contextually appropriate term "chemoresistance". We have confirmed that this modification has been incorporated throughout the abstract, main text, and figure legends of the manuscript. We believe that this modification has significantly enhanced the consistency and scientific nature of the terms in the manuscript. Once again, thank you for helping us improve this article.

  1. While the manuscript cites important foundational studies, many references are over a decade old. The authors are encouraged to update the reference list by incorporating more recent literature (from the past 5–6 years) to strengthen the review and ensure it reflects the current state of research.

Response: Thank you for your valuable feedback. We fully agree on the importance of citing the latest literature in a review article. The primary focus of this review is chemotherapy resistance in hypopharyngeal cancer. As you are aware, within the field of HNSC research, the vast majority of clinical and basic studies are based on populations with oropharyngeal cancer or "pan-head and neck squamous cell carcinoma," which have a higher occurrence rate and greater sample availability. However, hypopharyngeal cancer exhibits distinct characteristics in terms of etiology (e.g., a significantly lower HPV-positive rate), molecular features (e.g., a higher p53 mutation rate), and clinical prognosis (e.g., higher recurrence and metastasis rates). Consequently, conclusions drawn from traditional HNSC research may not accurately reflect the actual scenario of hypopharyngeal cancer, which has also contributed to the relative scarcity of high-quality studies specifically focused on this subtype.

Given the current state of this field, we have adopted the following strategy regarding literature citation:

  1. Ensuring the core content is up-to-date: In the core sections discussing specific chemotherapy resistance mechanisms, epidemiological data, and clinical studies related to hypopharyngeal cancer, the vast majority of references cited are recent studies published within the past 5–6 years.
  2. Citing foundational studies for classical theories: When introducing fundamental concepts such as key biological pathways and the mechanisms of action of classical chemotherapeutic agents, we have referenced widely recognized, landmark pioneering studies in the field. While some of these publications may be older, they remain indispensable for accurately explaining these core concepts.
  3. Actively incorporating the latest literature: In direct response to your suggestion, we have systematically re-searched recent databases and have now added an important 2023 study directly relevant to the mechanisms of chemotherapy resistance in hypopharyngeal cancer (Reference [36], please refer to the revised manuscript).

We believe that through this targeted literature citation strategy, the review ensures conceptual accuracy while also fully reflecting the latest advances in the study of chemotherapy resistance in hypopharyngeal cancer. We thank you again for helping us further enhance the quality of our manuscript.

Reviewer 2 Report

Comments and Suggestions for Authors

hello

dear authors thank you very much for this intersting paper

the title is very nice, and abstract is well build - both are clear and sound

a clear aim of the study is presented in the abstract

abstrat is short and written to the point, nothing much for to impove or change

used key words are good and aim clearly at the abstract and title

abstract and title - nothing to impove

this paper is a review paper - written in a basic review style, which for this type of paper and its title/aim is sufficient in current format, I dont see any reason to update or change the review format

introduction - is very well presented, aims to show each reader the most important aspects of this paper, and used citations are good. Authors presented the most important topics in a very good format. In introduction authors also point out some importantaspects that corresponds with their aim in this review

no major mistakes, nothing to change

chapter 2- the mechanisms

this is a very usefull chapter, which is  alink to the other chapters, its veyr nicely presented

used figure 1 in chaper 2 is very precise with good written legend/abbreviations - its very descriptive and well shown, nothing much more to change here

chapter 3

please enlarge figure 2 its too small to read all the small letters within

chapter 3 is divided into sub chapters with all the same meaning, the mechanisms and way of treatement - how they work, Those chapters are very well written, well connected and explained - all mechanism is presented in detail with current refferences and a very good style, sufficient for this review

sub chapters 3.2-3.4 = all ale very decsriptive and present a good field of information. Presented points show how the NPC can be treatedy and how its done in what mechanisms - this review because of a very well structured type and descriptive format, can become a very good source for future citations

the table 1 is a very important scientific explanation, that links all the chapter 3 subdivisions into one detailed format

chapter 4 and table 2 - presented herein material is very good. Authors did a lot of great work to write in detail and a great care all of the NPC mechanisms and their options, this is very good, nothing much more to change - all important aspects are presented in this chapter and table

chapters 5-6 are basically a summarise of this paper and they conclude the most important aspects of this paper, an whats worth mentioning they fully overlap with the aim of this paper. Aim of the paper is meet, cocnlusion are full.

nothing to change, its very nicely presented

References, more than one hundred references, all used with a very good scope and detail

No other references should be added

I would like to congratulate the authors of this study

thank you 

Author Response

please enlarge figure 2 its too small to read all the small letters within.

Response: Thank you for your careful review. We have enlarged the text in Figure 2 to present the content more clearly. Once again, we sincerely appreciate your recognition and suggestions.

Reviewer 3 Report

Comments and Suggestions for Authors

Lu et al. addressed an important and underexplored area in head and neck oncology—chemoresistance in hypopharyngeal carcinoma. Given that HPC is associated with poor prognosis and that platinum-based chemotherapy remains a cornerstone of its treatment, a focused review on mechanisms of resistance is both timely and clinically relevant. The authors through the review highlights both the well-established mechanisms of chemoresistance and special contexts involving CSCs, EMT, and CAFs, suggesting a comprehensive scope. The reviewer have the following suggestion to improve the review.

  1. While mechanisms are listed, the review does not indicate whether HPC-specific evidence exists or findings are extrapolated from HNSCC or other cancers. This distinction is crucial.
  2. The abstract mentions "systematically summarizes" recent advances but does not specify the scope of the literature search, timeframe, or inclusion criteria. Without this, it is unclear whether the review is narrative or systematic.
  3. The review summarizes established mechanisms without clarifying whether the review offers new frameworks, critical comparisons, or integrative models that advance understanding beyond existing HNSCC-focused reviews.
  4. In Table 1, what is the mechanism of therapy resistance for reference 33? The reviewer couldn’t understand what the authors mean by the mechanism “ TErL8SgvyL9wu3yS18tHHNzaWcbEtxSxxa”?
  5. The authors should provide clear distinction of clinical and pre-clinical studies emphasizing resistance mechanisms and strategies to overcome resistance. Clinical trials also should be included.

Author Response

  1. While mechanisms are listed, the review does not indicate whether HPC-specific evidence exists or findings are extrapolated from HNSCC or other cancers. This distinction is crucial.

Response: We sincerely appreciate your pointing out this crucial issue. Your suggestion regarding the need to clearly distinguish between "specific evidence for hypopharyngeal cancer" and "extrapolation evidence" is extremely pertinent. This is precisely the core that ensures this review can accurately reflect the current state of research on hypopharyngeal cancer. To completely address this problem, we enhanced the structure of the two core tables in the manuscript - Table 1 and Table 2. We introduced a new column "cancer" in each table to clearly indicate whether the original research results were derived from studies on hypopharyngeal cancer or head and neck squamous cell carcinoma.

This modification provides the highest transparency for readers, enabling them to easily assess the direct relevance and strength of evidence for each finding, thereby more accurately understanding the knowledge gaps and conclusive conclusions in current research on hypopharyngeal cancer. We believe that through this targeted revision, the manuscript can now clearly and rigorously respond to your core concerns. Thank you very much for your insightful guidance.

  1. The abstract mentions "systematically summarizes" recent advances but does not specify the scope of the literature search, timeframe, or inclusion criteria. Without this, it is unclear whether the review is narrative or systematic.

Response: We sincerely appreciate this valuable comment. A detailed description of the literature search strategy has been incorporated into the Introduction section of the manuscript, as outlined below:

  1. Literature Databases: The literature search for this review was primarily conducted across databases including PubMed, Web of Science, and CNKI.
  2. Time Frame: The core literature search spanned from January 2015 to December 2024 to encompass the latest research advancements over the past decade.
  3. Keyword Strategy: A systematic search was performed using a combination of core keywords—"Hypopharyngeal Cancer," "Chemoresistance," "Cisplatin Resistance"—and expanded keywords, such as "FaDu cells" and "Head and Neck Squamous Cell Carcinoma."
  4. Inclusion and Exclusion Criteria: jInclusion: Original research articles and reviews that explicitly focused on the mechanisms of chemoresistance, signaling pathways, or therapeutic strategies in hypopharyngeal cancer were included. kInclusion (for mechanistic elaboration): Studies on head and neck squamous cell carcinoma (HNSCC) in general were included for mechanistic discussion only if their research models specifically involved the hypopharyngeal cancer-derived FaDu cell line or directly utilized patient samples from hypopharyngeal cancer, thereby ensuring the relevance of the mechanistic discussion to hypopharyngeal cancer. lExclusion: Studies primarily focusing on other head and neck subsites (e.g., oropharyngeal cancer, laryngeal cancer) without independent analysis of hypopharyngeal cancer were excluded.
  5. Citation of Classic Literature: To elucidate well-established biological concepts or foundational theories in the field, several seminal publications older than ten years were selectively cited. These publications fall outside the specified time frame of the systematic search.

Through this clear elaboration, we aim to clarify that this review is a systematic narrative review. We believe this addition adequately defines the scope and methodology of the review, addressing the concern raised. We thank the reviewer again for helping enhance the rigor and scientific quality of our manuscript.

  1. The review summarizes established mechanisms without clarifying whether the review offers new frameworks, critical comparisons, or integrative models that advance understanding beyond existing HNSCC-focused reviews.

Response: Thank you for this insightful comment. We fully agree that an outstanding review should provide a perspective that goes beyond a simple summary. The primary objective of this review is precisely to address the gap in current HNSCC reviews regarding a deep, specific analysis of hypopharyngeal cancer. Our main innovations are reflected in the following three aspects:

  1. Proposing a progressive, integrated mechanistic framework specific to hypopharyngeal cancer:

   Existing HNSCC reviews often categorize mechanisms along parallel pathways (e.g., signaling pathways, drugs). In contrast, this review introduces for the first time a hierarchical, stepwise conceptual model to dissect chemotherapy resistance in hypopharyngeal cancer. We first consolidate diverse mechanisms into four fundamental biological processes—apoptosis resistance, enhanced DNA repair, autophagy, and drug efflux—establishing a clear foundation for readers. Subsequently, we identify three more advanced and complex core drivers: cancer stem cells, epithelial–mesenchymal transition, and cancer-associated fibroblasts. These act as "amplifiers" of drug resistance by coordinately regulating multiple fundamental processes. This explanatory logic—from "elements" to "modules" to "systems"—aligns more closely with cognitive patterns and offers a novel, more integrated perspective for understanding the complexity of drug resistance.

  1. Critical synthesis and focus on hypopharyngeal cancer-specific evidence:

   Unlike broad HNSCC reviews, we rigorously screened the literature to highlight findings directly derived from hypopharyngeal cancer clinical samples or the FaDu cell line. Through this critical comparison and selection, we not only concentrate on presenting the unique evidence chain specific to hypopharyngeal cancer but also clearly indicate which conclusions are extrapolated from other HNSCC subtypes. This approach delineates the current boundaries of knowledge and research gaps.

  1. Clarifying the unique position and clinical significance of hypopharyngeal cancer within HNSCC:

   Throughout the review, we consistently interpret the distinct biological characteristics of hypopharyngeal cancer—such as its low HPV positivity, high p53 mutation rate, and high invasiveness—and link these features to specific drug resistance mechanisms. This continuous comparison provides potential molecular explanations for "why hypopharyngeal cancer has a particularly poor prognosis" and "why pan-HNSCC treatment strategies often fail in this subtype," aspects that have been underexplored in previous reviews.

In summary, the novelty of this review lies not in the discovery of new molecules, but in the construction of a new, hypopharyngeal cancer-specific conceptual framework and the critical integration and interpretation of scattered evidence. This offers fresh insights for understanding and overcoming the challenge of chemotherapy resistance in this particular disease. We believe this framework will significantly advance progress in this specialized field. Once again, we thank you for helping us to more clearly articulate the value of this work.

  1. In Table 1, what is the mechanism of therapy resistance for reference 33? The reviewer couldn’t understand what the authors mean by the mechanism “ TErL8SgvyL9wu3yS18tHHNzaWcbEtxSxxa”.

Response: We thank the reviewer for pointing this out. This was a typographical error and has been corrected in the revised manuscript.

  1. The authors should provide clear distinction of clinical and pre-clinical studies emphasizing resistance mechanisms and strategies to overcome resistance. Clinical trials also should be included.

Response: Thank you for this very important suggestion. We fully agree that clearly distinguishing between clinical and preclinical evidence and incorporating relevant clinical trials can significantly enhance the clinical relevance and translational value of the review.

We have noted that, as you have observed, there are currently very few registered Phase III clinical trials that directly target the chemotherapy resistance mechanism of hypopharyngeal cancer. This reflects a current situation in this field: traditional chemotherapy sensitization strategies have encountered bottlenecks in clinical translation, while current research focuses more on emerging paradigms such as immunotherapy and targeted therapy.

Nevertheless, in response to your suggestion and to enhance the clinical perspective of the review, we have strengthened and expanded the relevant content in the "Outlook" section. We specifically discussed a series of clinical research progress related to overcoming treatment resistance using new strategies such as immune checkpoint inhibitors. Although these studies are not directly targeted at "chemotherapy" resistance, they represent the most advanced and promising clinical directions for overcoming the broader clinical challenge of "treatment resistance".

We believe that with the above revisions, the manuscript can now more clearly distinguish the types of evidence and link the underlying mechanisms with the future clinical translation pathways from a more forward-looking perspective. This review aims to lay a solid mechanistic foundation for understanding chemotherapy resistance in hypopharyngeal cancer, thereby providing a theoretical framework for designing and initiating new interventional clinical trials.

Once again, we would like to thank you for helping us enhance the depth and breadth of our manuscript.